# Competing Spin Liquid Phases in the S=$\frac{1}{2}$ Heisenberg Model on the Kagome Lattice

Shenghan Jiang,[1] Panjin Kim,[2] Jung Hoon Han,[2] and Ying Ran[1]

[1]*Department of Physics, Boston College, Chestnut Hill, MA 02467*
[2]*Department of Physics, Sungkyunkwan University, Suwon 16419, Korea*
(Dated: October 21, 2016)

The properties of ground state of spin-$\frac{1}{2}$ kagome antiferromagnetic Heisenberg (KAFH) model have attracted considerable interest in the past few decades, and recent numerical simulations reported a spin liquid phase. The nature of the spin liquid phase remains unclear. For instance, the interplay between symmetries and $Z_2$ topological order leads to different types of $Z_2$ spin liquid phases. In this paper, we develop a numerical simulation method based on symmetric projected entangled-pair states (PEPS), which is generally applicable to strongly correlated model systems in two spatial dimensions. We then apply this method to study the nature of the ground state of the KAFH model. Our results are consistent with that the ground state is a $U(1)$ Dirac spin liquid rather than a $Z_2$ spin liquid.

*Introduction* - Quantum spin liquids (QSL) can be defined as zero-temperature quantum phases of spin systems in the absence of symmetry breaking. In the presence of translational symmetry, and if there are odd number of half-integer spins per unit cell, the Hastings-Oshikawa-Lieb-Schultz-Mattis theorem[1–3] indicates that a QSL is necessarily a nontrivial quantum phase beyond the Landau's paradigm. It has been pointed out that geometric frustration, strong quantum fluctuation and/or strong spin-orbit coupling may be helpful to realize a QSL in realistic spin systems[4,5].

The nearest neighbour (NN) spin-$\frac{1}{2}$ kagome antiferromagnetic Heisenberg (KAFH) model is a spin model with a strong geometric frustration. Despite the simple form of the model Hamiltonian, the nature of the ground state of this model has been a long-standing puzzle and attracted considerable interest in the past few decades[6–19]. In particular, recently, a series of numerical simulations find this ground state to be a QSL. However, the nature of the QSL is still under debate. While numerical simulations based on density matrix renormalization group (DMRG) techniques[20,21] report evidences of a gapped $Z_2$ QSL[13–15], state-of-the-art variational Monte Carlo simulations find the ground state to be a gapless U(1) Dirac spin liquid[16]. In addition, it is known that there are many different candidate $Z_2$ QSLs that may be realized in this model[22–26] due to the interplay between the symmetry and the $Z_2$ topological order — a phenomenon coined symmetry enriched topological(SET) phases. Consequently it is still unclear which one of these candidate $Z_2$ QSLs may be realized in this model.

The difficulty of the problem, to a large extent, is due to the lack of suitable theoretical/numerical techniques. In order to simulate even moderate system sizes of frustrated quantum spin systems like the KAFH model, one has to work with certain kinds of variational wavefunctions. The choice of variational wavefunctions often brings up the following dilemma: On the one hand, one would like to work with wavefunctions in specific universality classes so that the analytical understanding of the simulation is available. This is the philosophy behind most variational Monte Carlo simulations. On the other hand, in order to perform an unbiased simulation and to obtain accurate energetics, one hopes that the choice of the variational wavefunctions is as general as possible. For instance, DMRG simulations are based on the matrix product states (MPS)[27,28] — a quite general class of variational wavefunctions.

The problem is that the two desired features of the variational wavefunctions usually do not come together. For example, different candidate $Z_2$ QSLs in the KAFH model are characterized by the different symmetry fractionalization patterns on the anyon quasiparticle excitations[29–32]. It is highly nontrivial to extract such analytical understandings from a MPS[33], although the DMRG simulations based on MPS provide very good energetics. At the same time, the variational Monte Carlo simulations based on the $U(1)$ Dirac spin liquid state[16], although having very clear analytical understanding, may be questioned about their generality.

It would be very interesting to develop new variational simulation schemes, hopefully capturing both desired features. This is indeed one of the motivations of an earlier piece of work by us, where we particularly pay attention to symmetric tensor-network wavefunctions[25,34]. In two spatial dimensions, we are focusing on the symmetric projected entangled pair states (PEPS)[35–37], which are natural generalizations of MPS. It turns out that one can systematically classify general PEPS wavefunctions according to symmetry. Consequently one can obtain a finite number of classes of symmetric PEPS wavefunctions, and perform a variational simulation within each class separately. On the one hand, the analytical understanding of each class of symmetric PEPS wavefunction is available, which is related to, but not limited to, the symmetry fractionalization phenomenon. On the other hand, because the classification of symmetric PEPS is quite general, after variationally simulating different classes of symmetric PEPS, one is expected to have rather good energetics and nearly unbiased understanding of the quantum phase diagram.

In this work, we further develop the numerical simulation scheme based on symmetric PEPS, and apply

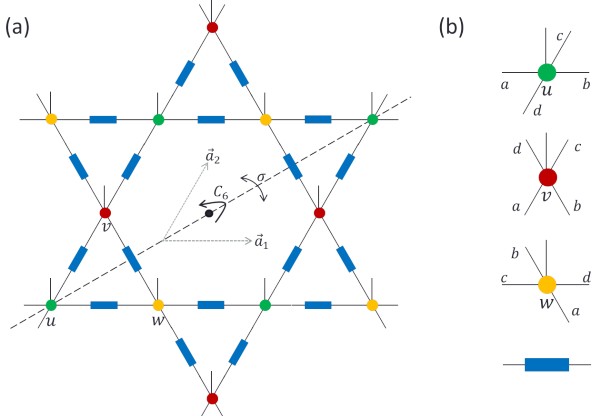

FIG. 1. (Color online) Schematic figures of the graphical representation of (a) the tensor network state on the kagome lattice (b) $u$, $v$ and $w$ comprised of four virtual legs and a physical leg describing Hilbert spaces of four virtual spins and a physical one respectively and bond tensor (blue rectangle) connecting neighboring virtual spins

it attempting to determine the nature of ground state of the KAFH model. The classification of symmetric PEPS[25,34] allows us to construct classes of generic PEPS wavefunctions for different $Z_2$ QSLs. We particularly focus on symmetric PEPS wavefunctions with bond dimension $D = 6$ and $D = 7$, and study the four $Z_2$ QSLs that can be realized by these bond dimensions. These four classes of symmetric PEPS correspond to Sachdev's $Q_1 = Q_2$ state, $Q_1 = -Q_2$ state and two other $\pi$-flux states. We perform variational simulations for the KAFH model for each class separately, and obtain four optimal energy densities. If the ground state is one of the four $Z_2$ QSLs, these optimal energy densities are expected to be significantly different, and the ground state is the $Z_2$ QSL with the lowest energy density.

However, surprisingly, we find that the optimal energy densities for both $Q_1 = Q_2$ state and $Q_1 = -Q_2$ state are nearly degenerate and comparable with the previously reported ground energy density of this model, while the two $\pi$-flux states have energy densities significantly higher. In fact, the most natural explanation for such a nearly degenerate energy density between the $Q_1 = Q_2$ state and $Q_1 = -Q_2$ state, without resorting to fine-tuning, is that the ground state is actually a $U(1)$ Dirac QSL. This is because both the $Q_1 = Q_2$ state and $Q_1 = -Q_2$ state can be viewed as descendent states from the same parent $U(1)$ Dirac QSL, and therefore can both be used to approximate the parent state. Consequently although we use $Z_2$ QSLs as trial wavefunctions, our results can be viewed as a supporting evidence of the $U(1)$ Dirac QSL.

*Spin-$\frac{1}{2}$ symmetric PEPS on kagome lattice* - The kagome PEPS and various notations for sites and bonds are shown in Fig. 1(a). To construct a spin-$\frac{1}{2}$ kagome PEPS, we associated every site/bond of the kagome lattice with a site/bond tensor. As shown in Fig. 1(b), a site tensor is formed by a physical leg which support a

physical spin-$\frac{1}{2}$, and four virtual legs, while a bond tensor is formed by two virtual legs. Every leg is associated with a specific local Hilbert space, and a tensor can be viewed as a quantum state in the Hilbert space of the tensor product of all its leg Hilbert spaces. The physical wavefunction is obtained by contracting all connected virtual legs of site tensors and bond tensors.

The classification of symmetric spin liquid phases on the kagome PEPS was obtained in Ref.25. Here, we briefly review the procedure and the result. The symmetry group of the spin-$\frac{1}{2}$ kagome system can be generated by translation symmetries $T_{1(2)}$, six-fold rotations about the center of the hexagon $C_6$, mirror reflection $\sigma$ along the dashed line in Fig.1(a), time-reversal symmetry $\mathcal{T}$, and spin rotation symmetry $U_{\theta\vec{n}}$. A global symmetry transformation $g$ induces a gauge transformation $W_g(x, y, s, i)$ on all internal legs of tensors. Here $(x, y, s)$ denotes the site position and $i$ labels the leg, as shown in Fig.1(b). Different spin liquid phases are characterized by gauge inequivalent symmetry transform rules $W_g$ on internal legs of the tensor network.

In our case, physical legs are spin-$\frac{1}{2}$'s, while internal legs support virtual spin representations. We can label a virtual Hilbert space as $\mathbb{V} = \bigoplus_{k=1}^{M}(\mathbb{D}_k \otimes \mathbb{V}_{S_k})$, where $\mathbb{V}_{S_k}$ supports spin $\vec{S}_k$ and $M$ denotes number of spin species, while $\mathbb{D}_k$ is the flavor space. The dimension of $\mathbb{V}$ is $D = \sum_{k=1}^{M} n_k(2S_k + 1)$. As shown in Ref.25, the global $2\pi$ spin rotation induces a special *pure gauge transformation* $\{J\}$, which leaves *every single tensor invariant* up to some phase factor. For instance, when $\mathbb{V} = 0 \oplus \frac{1}{2}$, $J = \text{diag}[1, -1, -1]$ on every internal leg. $\{J\}$ together with the identity action form a $Z_2$ invariant gauge group (IGG), which is related to the $Z_2$ toric code topological order.

The $Z_2$ IGG will enter tensor equations for symmetries and enrich the classification. Briefly speaking, $W_g$, which is the symmetry action on internal legs, satisfies group multiplication rules up to an IGG element (either trivial or nontrivial) as well as a phase factor. Given the $Z_2$ IGG and global symmetries of the model, one obtains 32 inequivalent classes, which are characterized by five $Z_2$ indices: $\eta_{12}, \eta_{C_6}, \eta_\sigma$ and $\chi_\sigma, \chi_\mathcal{T}$. Here $\eta$'s label $Z_2$ IGG elements, which characterize symmetry fractionalizations of spinon $e$-particles, while $\chi$'s are phase factor $\pm 1$, which are related to "weak SPT" indices. For example, $\eta_{12} = I/J$ corresponds to zero-flux/$\pi$-flux spin liquids in the Schwinger boson language.

As listed in Appendix A, for all classes, we solve the symmetry transformation rules $W_g$ for arbitrary $D$ by fixing gauge. The fact that tensors are invariant under symmetry actions on both physical legs and internal legs imposes constraints on the Hilbert space of local tensors. Tensors of different classes live in different constraint sub-Hilbert spaces. Here, we focus on two cases: $D = 6$ with virtual spins $0 \oplus \frac{1}{2} \oplus 1$ and $D = 7$ with virtual spins $0 \oplus 0 \oplus \frac{1}{2} \oplus 1$. Only 4 of the 32 classes can be realized in these two cases, which are fully characterized by two indices $\eta_{12}$ and $\eta_{C_6}$ while other indices are fixed as

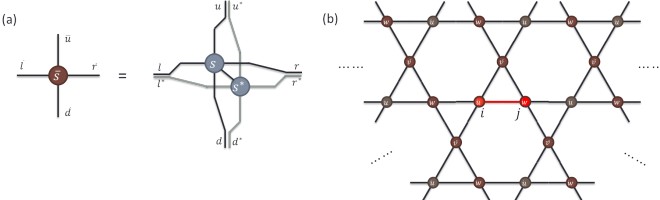

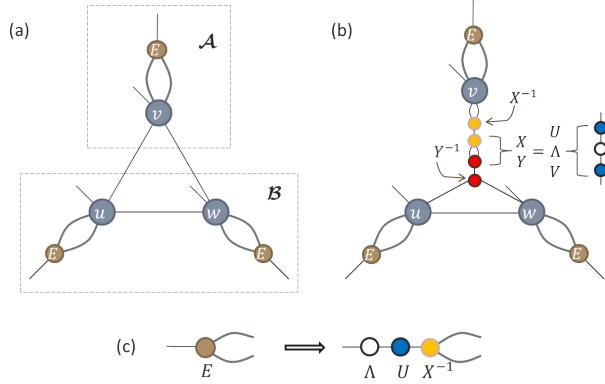

FIG. 2. (Color online) (a) A double layer site tensor from contraction of physical legs. (b) Graphic representation of $\langle\psi|h_{ij}|\psi\rangle$, where $h_{ij}$ acts on the red sites and bond.

FIG. 3. (Color online) (a) Site tensors in a unit cell with environment matrix $E$ are decomposed to $\mathcal{A}$ and $\mathcal{B}$. (b) Insert $X^{-1}XYY^{-1}$ projectors between $\mathcal{A}$ and $\mathcal{B}$, and replace $X \cdot Y$ with its singular value decomposition $U\Lambda V$. (c) Update the environment matrix $E$ to be $\Lambda UX^{-1}$.

$\eta_\sigma = J$ and $\chi_\sigma = \chi_\mathcal{T} = 1$. These four classes happen to contain four types of NN RVB states[23,38–41] as representative wavefunctions. For $D = 6$, in the absence of any symmetry, the Hilbert space of the site tensor would be $2 \cdot 6^4 = 2592$-dimensional. After implementing all symmetries, the constrained sub-Hilbert space $\mathbb{V}_{site}$ of a site tensor turns out to be only 19-dimensional for all four classes, which significantly reduces the number of variational parameters. (For $D = 7$, the symmetry-constrained sub-Hilbert space $\mathbb{V}_{site}$ is 43-dimensional for all four classes.)

*Symmetric iPEPS algorithm* - Given generic tensor wavefunctions for all classes, our goal is to find the optimal PEPS wavefunction for each class, which minimize $\langle h_{ij}\rangle = \langle\Psi|h_{ij}|\Psi\rangle$, where $h_{ij}$ is the local Hamiltonian acting on two neighbouring sites $i, j$. In NN KAFH, $h_{ij} = \vec{S}_i \cdot \vec{S}_j$.

As shown in Fig.2(b), $\langle h_{ij}\rangle$ is calculated by contracting all legs of a double layer PEPS. Notice that we have already absorbed bond tensors to neighbouring site tensors for convenience. The bond dimension of the double layer PEPS is $D^2$, and it is generally impossible to get the exact result of tensor contraction. The key point of the iPEPS algorithm is to find a reasonable approximation for the environment tensor around site $i, j$. The algorithm is divided into two parts: optimization and measurement. In the following, we will describe these two parts separately.

**Optimization.** In this paper, we apply a modified simple update algorithm[42] to optimize the wavefunction. As shown in Fig.3, for the simple update method, the environment tensors of three sites are approximated by the direct product of matrices $E$. Here, all sites share the same matrix $E$ due to lattice symmetries.

The algorithm to obtain $E$ is described in the following:

1. First, we define the local wavefunction $|\psi\rangle$ as contracting single layer site tensors in one unit cell with initial environment matrix $E$. As shown in Fig.3(a), we can decompose $|\psi\rangle$ as $|\psi\rangle = \sum_\alpha |\phi_\alpha^\mathcal{A}\rangle \otimes |\phi_\alpha^\mathcal{B}\rangle$, where $\alpha$ labels virtual states living in the tensor product space of leg $uv$ and $vw$.

2. Define $M_{\alpha\alpha'} = \langle\phi_\alpha^\mathcal{A}|\phi_{\alpha'}^\mathcal{A}\rangle$. Then, $M$ is a hermitian matrix, and can be decomposed as $M = (X^\mathrm{T})^\dagger \cdot X^\mathrm{T}$. The decomposition can be sped up a lot by implementing spin rotation symmetries. Then, $|e_\alpha\rangle \equiv$

$(X^{-1})_{\alpha\alpha'}|\phi_{\alpha'}\rangle$ form an orthonormal set. Similar analysis on $\mathcal{B}$ leads to an orthonormal set $\{|f_\beta\rangle\}$. As shown in Fig.3(b), $|\psi\rangle = \sum_{\alpha\beta} X_{\alpha\gamma}Y_{\gamma\beta}|e_\alpha\rangle \otimes |f_\beta\rangle$. We then perform singular value decomposition: $X \cdot Y = U\Lambda V$, where $\Lambda$ encodes the entanglement information of $|\psi\rangle$.

3. Update one environment matrix $E$ to be $E \to \Lambda UX^{-1}$, as shown in Fig.3(c), and then use spatial symmetry transformation rules in Appendix A to generate all environment matrices at different spatial positions.

4. Repeat the above procedure until $\Lambda$ converges.

Given an arbitrary PEPS wavefunction $|\Psi\rangle$ belonging to some spin liquid class, say, class A, we are able to efficiently measure the approximate "energy density" $\langle h_{ij}\rangle_{su}$ using the converged environment matrix $E$. We then implement standard minimization algorithm, for instance, the conjugate gradient method, to search for the optimal wavefunction in the constraint sub-Hilbert space of class A, which minimize $\langle h_{ij}\rangle_{su}$. Notice that the major advantages of this simple-update algorithm are its stability and speed, although the approximation introduced by the direct-product-environment $E$ is not well under control. In order to control the approximation in the environment tensor, other algorithms like full-update[43] need to be used, which we leave as a topic of future studies.

**Measurement.** By implementing the optimization algorithm to all four classes, we obtain optimal wavefunctions for these classes. We then measure the energy density of each optimal wavefunction as accurately as we can. We mainly use variational Monte Carlo combined with tensor entanglement renormalization method (VMC-TERG)[44] to measure the energy density on a 192-site finite-size sample. (The energy density measurement based on iTEBD algorithm[45,46] is also per-

TABLE I. The optimal energy per site $E$ for the four promising classes with virtual bond dimension $D = 6$ and $D = 7$. Error bars here are due to fitting errors.

| Classes | $D = 6$ | $D = 7$ |
|---|---|---|
| zero-flux I | -0.4354(2) | -0.4366(3) |
| zero-flux II | -0.4351(6) | -0.4365(5) |
| $\pi$-flux I | -0.4293(5) | -0.4313(7) |
| $\pi$-flux II | -0.4296(8) | -0.4227(4) |

formed as a complementary check. See Appendix B for details.) VMC-TERG is a single-layer algorithm in which one has to approximate the tensor-contraction by keeping a finite bond-dimension $D_{cut}$ during the real-space tensor renormalization. Namely, a finite $D_{cut}$ would introduce approximation and a scaling analysis with respect to $D_{cut}$ is usually necessary. However, we would like to emphasize that despite having approximation for the tensor-contraction, for any given $D_{cut}$, the energy measurement by VMC-TERG is *variational*. This sharp variational meaning of the VMC-TERG algorithm is one of its major advantage comparing with other algorithms.

*Result* - We perform the above algorithm to the four promising classes with virtual bond dimension $D = 6$ and $D = 7$. Results measured by VMC-TERG are presented in Fig.4. Energy densities of optimized wavefunctions are measured in the $8 \times 8 \times 3$ kagome lattice, and the scaling over $D_{cut}$ is applied. We fit energy densities as power law functions of $D_{cut}$: $E \sim \frac{1}{D_{cut}^{\alpha}}, 1 \leq \alpha \leq 2$, where fitting parameter $\alpha$ is chosen to have the best fitting quality[47]. Energy densities obtained from extrapolation to infinite $D_{cut}$ are presented in Table I. We warn the readers that this type of extrapolation-schemes is only empirically justified[44,48].

As shown in Fig.4 and Table I, energy densities of the two zero-flux classes are significantly lower than those of the two $\pi$-flux classes, which indicates the ground state of NN KAFH should be a zero-flux spin liquid. However, these two zero-flux classes have degenerate optimal energy density for both $D = 6$ and $D = 7$ within error bar. It appears that our method fails to determine the correct class of the $Z_2$ spin liquid for the NN KAFH model.

In fact, the most natural way to interpret the energy degeneracy is that the ground state is actually a $U(1)$ Dirac spin liquid[9,16,49] rather than a gapped $Z_2$ spin liquid. To justify this statement, we note that the $U(1)$ Dirac spin liquid is the "parent class" of these two zero-flux $Z_2$ spin liquids. One way to see this is to go to the Abrikosov fermion language[38,50,51], in which the $U(1)$ Dirac spin liquid is described by gapless fermionic spinons coupling to the internal $U(1)$ gauge field. By adding pairings of fermionic spinons, the $U(1)$ gauge field will be Higgsed to $Z_2$, leading to $Z_2$ spin liquids. Patterns of pairing are constrained by lattice symmetries, and different pairing patterns give different $Z_2$ spin liquids.

It turns out that these two zero-flux classes are exactly the neighboring phases of the same $U(1)$ Dirac spin

liquid[24], while the two $\pi$-flux states are *not* neighboring phases of the $U(1)$ Dirac spin liquid.[52] Consequently, any state belonging to the $U(1)$ Dirac spin liquid can be approximated by wavefunctions of these two descendant zero-flux $Z_2$ spin liquids classes by turning on very small pairing. This would naturally lead to the optimal energy degeneracy obtained by the two zero-flux classes of symmetric PEPS, without having to resort to fine-tuning.

The optimal energy density measured here on the 192-site sample is comparable to the thermodynamic-limit energy density reported in a recent tensor-network-based work Ref.53, and is slightly higher than the estimated thermodynamic-limit energy density obtained from DMRG[13]. We expect that the optimal variational energy can be further improved by implementing more accurate optimization methods, such as the fast full update algorithm[43].

*Discussion and Conclusion* - We demonstrate a new variational numerical simulation scheme based on symmetric PEPS wavefunctions. Although we study the particular KAFH model in this paper, the classification and simulation of symmetric PEPS wavefunctions are generally applicable to other correlated quantum systems. The main advantage of this scheme is that two desired features of variational simulations are both realized. First, the systematic classifications and constructions of generic PEPS wavefunctions allow one to simulate the quantum phase without losing generality and obtain accurate energetics, which can be comparable with the energetics of other state-of-the-art variational methods. Second, despite being general, sharp analytical understandings for each class of symmetric PEPS wavefunctions are available.

In particular, we simulate four promising candidate spin liquids on the KAFH model. Two distinct zero-flux $Z_2$ QSLs give nearly degenerate optimal energy density which is comparable with the ground state energy density reported using other methods. The most natural explanation for this degeneracy is that the ground state is actually the $U(1)$ Dirac spin liquid, which is the parent phase of both classes.

It is also informative to compare our simulation with previous parton-based variational studies on the two zero-flux $Z_2$ QSLs. Ref.54 reported the variational Monte Carlo simulations based on Guzwiller-projected Schwinger-boson states, and it was found that the $Q_1 = Q_2$ state has an energy density significantly lower than that of the $Q_1 = -Q_2$ state. Although the Guzwiller-projected Schwinger-boson states are in the same universality classes as the two symmetric PEPS classes studied here, the energetics performance of the PEPS wavefunctions are much better. This can be intuitively understood as follows. The tunable variational parameters in parton-based wavefunctions quickly become long-ranged in the real space as one increases the number of parameters, which would not improve energetics — a short-range property of the wavefunctions. However, the tunable variational parameters in symmetric PEPS wave-

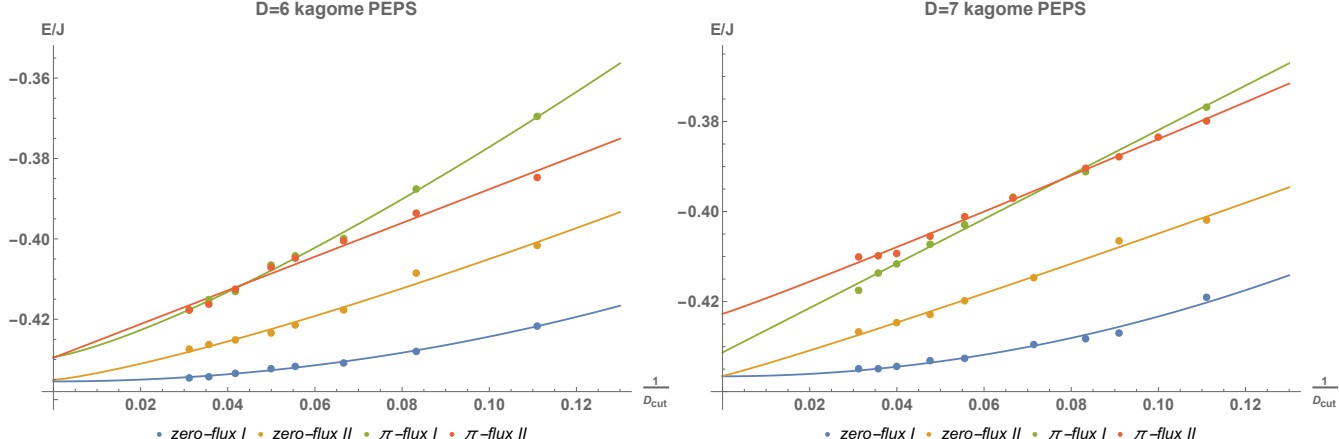

FIG. 4. (Color online) Numerical results on the spin-$\frac{1}{2}$ kagome PEPS for optimal energy densities of four promising classes with $D = 6$ (left) and $D = 7$ (right) measured by VMC combined with TERG on the $8 \times 8 \times 3$ kagome lattice. The zero-flux I/II class is the $Q_1 = Q_2/Q_1 = -Q_2$ class in Ref.22. Error bars are smaller than size of data points, so are not displayed. Power law functions $E \sim \frac{1}{D_{cut}^\alpha}$ ($1 \leq \alpha \leq 2$) are used to fit the data.

functions are directly enlarging the local Hilbert space for a local tensor, which can significantly improve energetics.

We thank Ling Wang, E. Miles Stoudenmire and Patrick Lee for helpful discussions, and Yin-Chen He for sharing his unpublished results on this model using DMRG techniques. The PEPS calculations were based on the ITensor library, http://itensor.org/. This study is supported by the Alfred P. Sloan fellowship and National Science Foundation under Grant No. DMR-1151440. We thank Boston College Research Service for providing the computing facilities where the numerical simulations were performed.

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

## Appendix A: Symmetry transformation rules and constrained sub-Hilbert spaces for the symmetric kagome PEPS classes

### 1. The symmetry group for KAFH

As shown in Fig.(1), we label the three lattice sites in each unit cell with sublattice index $\{s = u, v, w\}$. Further, we specify the virtual index $\{i = a, b, c, d\}$ of a given site. We choose Bravais unit vector as $\vec{a}_1 = \hat{x}$ and $\vec{a}_2 = \frac{1}{2}(\hat{x} + \sqrt{3}\hat{y})$. Thus, we are able to specify the virtual degrees of freedom of site tensors as $(x, y, s, i)$. The symmetry group of such a two-dimensional kagome lattice is

generated by the following operations

$$
\begin{aligned}
T_1 &: (x, y, s, i) \rightarrow (x + 1, y, s, i), \\
T_2 &: (x, y, s, i) \rightarrow (x, y + 1, s, i), \\
\sigma &: (x, y, u, i) \rightarrow (y, x, u, i_{\sigma 1}), \\
&\quad (x, y, v, i) \rightarrow (y, x, w, i_{\sigma 2}), \\
&\quad (x, y, w, i) \rightarrow (y, x, v, i_{\sigma 2}), \\
C_6 &: (x, y, u, i) \rightarrow (-y + 1, x + y - 1, v, i), \\
&\quad (x, y, v, i) \rightarrow (-y, x + y, w, i). \\
&\quad (x, y, w, i) \rightarrow (-y + 1, x + y, u, i_{C_6}).
\end{aligned}
\tag{A1}
$$

together with time reversal $\mathcal{T}$. Here,

$$
\begin{aligned}
\{a_{\sigma 1}, b_{\sigma 1}, c_{\sigma 1}, d_{\sigma 1}\} &= \{d, c, b, a\} \\
\{a_{\sigma 2}, b_{\sigma 2}, c_{\sigma 2}, d_{\sigma 2}\} &= \{c, d, a, b\} \\
\{a_{C_6}, b_{C_6}, c_{C_6}, d_{C_6}\} &= \{b, a, d, c\}
\end{aligned}
$$

The symmetry group of a kagome lattice is defined by the following algebraic relations between its generators:

$$
\begin{aligned}
T_2^{-1} T_1^{-1} T_2 T_1 &= \mathrm{e}, \\
\sigma^{-1} T_1^{-1} \sigma T_2 &= \mathrm{e}, \\
\sigma^{-1} T_2^{-1} \sigma T_1 &= \mathrm{e}, \\
C_6^{-1} T_2^{-1} C_6 T_1 &= \mathrm{e}, \\
C_6^{-1} T_2^{-1} T_1 C_6 T_2 &= \mathrm{e}, \\
\sigma^{-1} C_6 \sigma C_6 &= \mathrm{e}, \\
C_6^6 = \sigma^2 = \mathcal{T}^2 &= \mathrm{e}, \\
g^{-1} \mathcal{T}^{-1} g \mathcal{T} = \mathrm{e}, \forall g &= T_{1,2}, \sigma, C_6
\end{aligned}
\tag{A2}
$$

where e stands for the identity element in the symmetry group.

Further, consider system with spin rotation symmetry operator $R_{\theta\vec{n}}$, which means spin rotation about axis $\vec{n}$ through angle $\theta$. We mainly consider half-integer spins ($SU(2)$ symmetry) in this paper. The spin rotation symmetry commutes with all lattice symmetries as well as time reversal symmetry:

$$
g^{-1} R_{\theta\vec{n}}^{-1} g R_{\theta\vec{n}} = \mathrm{e}, \forall g = T_{1,2}, \sigma, C_6, \mathcal{T}
\tag{A3}
$$

$$
\tag{A4}
$$

### 2. Symmetry transformation rules on internal legs

There are $2^5 = 32$ symmetric PEPS classes, labeled by five $Z_2$ indices: $\{\eta_{12}, \eta_{C_6}, \eta_\sigma, \chi_\sigma, \chi_\mathcal{T}\}$, where $\eta = I/J$ and $\mu = \pm 1$. We choose $J$ to be the direct sum of $I_{D_1}$ for the integer spin subspace and $-I_{D_2}$ for the half-integer spin subspace by fixing gauge.

As shown in Ref.25, symmetry transformation rules

$W_g$ on internal legs can be represented as:

$$W_{T_1}(x, y, s, i) = \eta_{12}^y,$$
$$W_{T_2}(x, y, s, i) = I,$$
$$W_{C_6}(x, y, u, i) = \eta_{12}^{xy + \frac{1}{2}x(x+1) + x + y} w_{C_6}(u, i),$$
$$W_{C_6}(x, y, v, i) = \eta_{12}^{xy + \frac{1}{2}x(x+1) + x + y},$$
$$W_{C_6}(x, y, w, i) = \eta_{12}^{xy + \frac{1}{2}x(x+1)},$$
$$W_\sigma(x, y, s, i) = \eta_{12}^{x + y + xy} w_\sigma(s, i),$$
$$W_\mathcal{T}(x, y, s, i) = w_\mathcal{T}(s, i),$$
$$W_{\theta\vec{n}}(x, y, s, i) = \bigoplus_i (I_{n_i} \otimes e^{i\theta\vec{n}\cdot\vec{S}_i}). \tag{A5}$$

For the rotation transformation $w_{C_6}(u, i)$, we have

$$w_{C_6}(u, a) = w_{C_6}(u, c) = I,$$
$$w_{C_6}(u, b) = w_{C_6}(u, d) = \eta_{12}\eta_{C_6}, \tag{A6}$$

For the reflection transformation $w_\sigma(s, i)$, we have

$$w_\sigma(u, a) = I, \quad w_\sigma(u, b) = \chi_\sigma\eta_{12}\eta_{C_6},$$
$$w_\sigma(u, c) = \chi_\sigma\eta_{12}\eta_{C_6}\eta_\sigma, \quad w_\sigma(u, d) = \eta_\sigma;$$
$$w_\sigma(v, a) = \eta_{12}, \quad w_\sigma(v, b) = \chi_\sigma\eta_{12},$$
$$w_\sigma(v, c) = \eta_{C_6}\eta_\sigma, \quad w_\sigma(v, d) = \chi_\sigma\eta_{C_6}\eta_\sigma;$$
$$w_\sigma(w, a) = \chi_\sigma\eta_{C_6}, \quad w_\sigma(w, b) = \eta_{C_6},$$
$$w_\sigma(w, c) = \eta_{12}\eta_\sigma, \quad w_\sigma(w, d) = \chi_\sigma\eta_{12}\eta_\sigma; \tag{A7}$$

And for the time reversal transformation $w_\mathcal{T}$, we have

$$w_\mathcal{T}(u, a) = w_\mathcal{T}, \quad w_\mathcal{T}(u, b) = \eta_{12}\eta_{C_6}w_\mathcal{T},$$
$$w_\mathcal{T}(u, c) = \eta_{12}\eta_{C_6}\eta_\sigma w_\mathcal{T}, \quad w_\mathcal{T}(u, d) = \eta_\sigma w_\mathcal{T};$$
$$w_\mathcal{T}(v, a) = \eta_{12}\eta_{C_6}w_\mathcal{T}, \quad w_\mathcal{T}(v, b) = w_\mathcal{T},$$
$$w_\mathcal{T}(v, c) = \eta_\sigma w_\mathcal{T}, \quad w_\mathcal{T}(v, d) = \eta_{12}\eta_{C_6}\eta_\sigma w_\mathcal{T};$$
$$w_\mathcal{T}(w, a) = w_\mathcal{T}, \quad w_\mathcal{T}(w, b) = \eta_{12}\eta_{C_6}w_\mathcal{T},$$
$$w_\mathcal{T}(w, c) = \eta_{12}\eta_{C_6}\eta_\sigma w_\mathcal{T}, \quad w_\mathcal{T}(w, d) = \eta_\sigma w_\mathcal{T}; \tag{A8}$$

where

$$w_\mathcal{T} = \begin{cases} \bigoplus_i (I_{n_i} \otimes e^{i\pi S_i^y}) & \text{if } \chi_\mathcal{T} = 1 \\ \bigoplus_i (\Omega_{n_i} \otimes e^{i\pi S_i^y}) & \text{if } \chi_\mathcal{T} = -1 \end{cases} \tag{A9}$$

Here $n_i$ is dimension of the extra degeneracy associated with spin-$S_i$. Namely, the total degeneracy for spin-$S_i$ living on one virtual leg equals $n_i \times (2S_i + 1)$. We have the virtual bond dimension

$$D = \sum_i n_i(2S_i + 1) \tag{A10}$$

And, $\Omega_{n_i} = i\sigma_y \otimes I_{n_i/2}$ is a $n_i$ dimensional antisymmetric matrix.

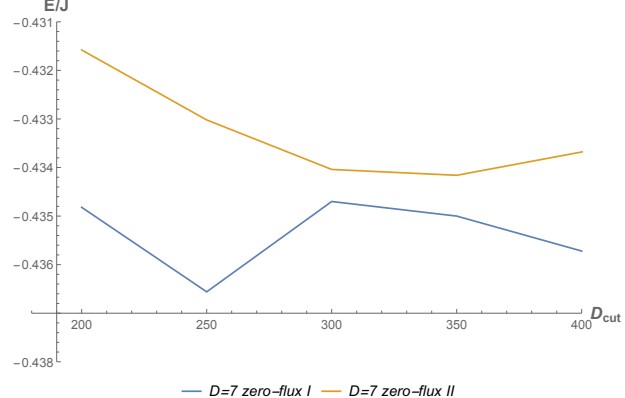

FIG. 5. (Color online) Optimal energy densities of two zero flux classes measured by the iTEBD method.

For $\Theta_R$'s, we have

$$\Theta_{T_1}(x, y, s) = \mu_{12}^y, \quad \Theta_{T_2}(x, y, s) = 1,$$
$$\Theta_{C_6}(x, y, u) = \mu_{12}^{xy + \frac{1}{2}x(x+1) + x + y}\Theta_{C_6}(u),$$
$$\Theta_{C_6}(x, y, v) = \mu_{12}^{xy + \frac{1}{2}x(x+1) + x + y},$$
$$\Theta_{C_6}(x, y, w) = \mu_{12}^{xy + \frac{1}{2}x(x+1)},$$
$$\Theta_\sigma(x, y, s) = \mu_{12}^{x + y + xy}\Theta_\sigma(s),$$
$$\Theta_\mathcal{T}(x, y, u/w) = 1, \quad \Theta_\mathcal{T}(x, y, v) = \mu_{12}\mu_{C_6},$$
$$\Theta_{\theta\vec{n}} = 1, \tag{A11}$$

where

$$\Theta_{C_6}(u) = (\mu_{12}\mu_{C_6})^{\frac{1}{2}};$$
$$\Theta_\sigma(u) = (\mu_\sigma)^{\frac{1}{2}};$$
$$\Theta_\sigma(v) = \mu_{C_6}\Theta_{C_6}(u)\Theta_\sigma(u);$$
$$\Theta_\sigma(w) = \mu_\sigma\mu_{C_6}(\Theta_{C_6}(u)\Theta_\sigma(u))^{-1}. \tag{A12}$$

## Appendix B: Optimal energies measured by iTEBD

We use the iTEBD method[45] to measure the energy densities of optimal wavefunctions belonging to two zero flux classes with $D = 7$. As shown in Fig. 5, the energy densities are still fluctuating up to $D_{cut} = 400$, and it is hard to see the trend for larger $D_{cut}$. However, these results are in agreement with the VMC-TERG result if one intuitively treats the fluctuation ranges as error bars.