# Peer review of "Competing Spin Liquid Phases in the S=$\frac{1}{2}$ Heisenberg Model on the Kagome Lattice"

_SciPost Physics_

## Round 2 · Referee Report · Didier Poilblanc · 2019-5-17

Strengths

The Kagome HAFM is one of the hardest problem in frustrated quantum magnetism. The authors use state-of-the-art, recently developed, reliable tensor network (TN) techniques to investigate the nature of the spin liquid ground-state, obtaining convincing results in favor of a critical Dirac spin liquid.

Weaknesses

Naively, one may think this paper is a bit outdated, considering the fact that more recent work on the topic, reaching similar conclusions, have been published as e.g. Liao et al., Phys. Rev. Lett. 118, 137202 (2017) by the group of Professor Tao Xiang (see comment below).

Report

Considering the two remarks above, my overall opinion about the paper is very positive and I think it should be published because:

i) it was initially submitted on arXiv at the same time (even a week before) as the above mentioned PRL, while drawing similar strong conclusions.

ii) it follows a different route, using state-of-the-art tensor symmetry analysis, enabling to construct fully SU(2)-symmetric ansatz while in the non-symmetric TN version a spurious finite 120-degrees magnetic order at all finite D (the tensor bond dimension) values appears.

---

## Round 2 · Referee Report · Anonymous · 2019-5-24

Strengths

1- New numerical methods brought to bear on an old problem
2- Compelling evidence for the U(1) spin liquid being at least very nearly degenerate with the true ground state

Weaknesses

1- Relies on empirical extrapolation of energy in the truncation used to compute energies, which may be unreliable.

Report

The authors revisit the Kagome Heisenberg antiferromagnet, and study it using PEPS variational wave functions restricted to certain gauge symmetry classes, and thus certain classes of spin liquid phases. They find that two possible distinct Z2 spin liquids give nearly degenerate energies, and thus conclude that their shared "parent phase", the U(1) spin liquid, is the actual ground state of the system.

Overall, I find the paper well-written and relevant. The numerics seem to be carried out carefully; while some of the methods are rather new and their performance poorly understood, the authors point out quite explicitly that this is the case (for example that the energy extrapolation used in Fig. 4 is purely empirical).

Indeed, this extrapolation seems crucial to the paper. It is quite notable that for a given D_cut, the two states are not even close to degenerate, and only after extrapolation, they become degenerate. I would ask that the authors include exactly what fit parameters were used, which will help assessment and later reproducibility of the results. Also, judging from the plot, the exponents for the two lowest states must be quite different - do the authors have an explanation of why that would be the case? Ideally, the authors could also perform some sort of robustness analysis of their fit.

With the small modifications suggested above, the paper can be published.

Requested changes

1- List fit coefficients.
2- Ideally perform robustness analysis of the fit.

---

## Round 3 · Author Response

We have made the requested modifications. We list the fitting parameters for energies(exponent) on Fig. 4. We also add an appendix to do robustness analysis for the two zero-flux states. The fitting for zero-flux I phase is more reliable given the almost energy convergence at large D_cut. For zero-flux II phase, we present another fitting scheme, which gives consistent energies at infinite D_cut.

---

## Round 3 · List of Changes

1. add fitting parameters on Fig. 4
2. add another appendix performing robustness analysis.

Resubmission 1610.02024v3 on 12 June 2019
Submission 1610.02024v2 on 24 April 2019

---

## Editorial Decision

ontology_/_topics